# Novel perspective on a conventional technique: Impact of ultra-low temperature on bacterial viability and protein extraction

**Aditya Sarnaik[1], Apurv Mhatre[1], Muhammad Faisal[1,2], Dylan Smith[1], Ryan Davis[3], Arul M. Varman**◉[1]*

**1** Department of Chemical Engineering, School for Engineering of Matter, Transport, and Energy, Arizona State University, Tempe, AZ, United States of America, **2** University Institute of Biochemistry and Biotechnology, PMAS-Arid Agriculture University Rawalpindi, Rawalpindi, Pakistan, **3** Bioresource and Environmental Security, Sandia National Laboratories, Livermore, CA, United States of America

* Arul.M.Varman@asu.edu

**Data Availability Statement:** All relevant data are within the manuscript and its Supporting Information files.

## Abstract

Ultra-low temperature (ULT) storage of microbial biomass is routinely practiced in biological laboratories. However, there is very little insight regarding the effects of biomass storage at ULT and the structure of the cell envelope, on cell viability. Eventually, these aspects influence bacterial cell lysis which is one of the critical steps for biomolecular extraction, especially protein extraction. Therefore, we studied the effects of ULT-storage (-80°C) on three different bacterial platforms: *Escherichia coli*, *Bacillus subtilis* and the cyanobacterium *Synechocystis* sp. PCC 6803. By using a propidium iodide assay and a modified MTT assay we determined the impact of ULT storage on cellular viability. Subsequently, the protein extraction efficiency was determined by analyzing the amount of protein released following the storage. The results successfully established that longer the ULT-storage time lower is the cell viability and larger is the protein extraction efficiency. Interestingly, *E. coli* and *B. subtilis* exhibited significant reduction in cell viability over *Synechocystis* 6803. This indicates that the cell membrane structure and composition may play a major role on cell viability in ULT storage. Interestingly, *E. coli* exhibited concomitant increase in cell lysis efficiency resulting in a 4.5-fold increase (from 109 µg/ml of protein on day 0 to 464 µg/ml of protein on day 2) in the extracted protein titer following ULT storage. Furthermore, our investigations confirmed that the protein function, tested through the extraction of fluorescent proteins from cells stored at ULT, remained unaltered. These results established the plausibility of using ULT storage to improve protein extraction efficiency. Towards this, the impact of shorter ULT storage time was investigated to make the strategy more time efficient to be adopted into protocols. Interestingly, *E. coli* transformants expressing mCherry yielded 2.7-fold increase (93 µg/mL to 254 µg/mL) after 10 mins, while 4-fold increase (380 µg/mL) after 120 mins of ULT storage in the extracted soluble protein. We thereby substantiate that: (1) the storage time of bacterial cells in -80°C affect cell viability and can alter protein extraction efficiency; and (2) exercising a simple ULT-storage prior to bacterial cell lysis can improve the desired protein yield without impacting its function.

**Funding:** AMV acknowledges start-up funds from the School for Engineering of Matter, Transport and Energy at Arizona State University and the awards #2184871 and #1871463 received from Sandia National Laboratories. This research was also supported in parts by the BioEnergy Technology Office (BETO), U.S. Department of Energy, through support of RWD, under agreement #26336. The funders had no role in study design, data collection and analysis, decision to publish, or preparation of the manuscript.

**Competing interests:** The authors have declared that no competing interests exist.

## Introduction

Microbial genetic engineering has revolutionized the field of biotechnology for improving the production of economically viable inherent metabolite/s, heterologous biomolecules, and value-added chemicals [1–4]. Ultra-low temperature (ULT) storage of the recombinant cells and harvested microbial biomass is a routine exercise in biological laboratories before subsequent sample processing; but its impact on cell lysis efficiency and proteins of interest has been minimally reported. On the other hand, traditional storage techniques including freeze drying, and rapid chilling, which involve a freezing step are known to negatively impact bacterial survival over long duration [5–7]. Additionally, freeze-thaw techniques are widely used for bacterial cell disruption. These recurring observations indicate the plausible derogatory impact of ULT on bacterial cell membrane and cell viability. Furthermore, it is important to investigate the impact of cell composition and the duration of ULT-storage to identify their combined effect on cell lysis efficiency. With this background, we sought to systematically investigate and quantify the impact of ULT storage on bacterial platforms possessing distinct cell envelope structure and composition.

Cell lysis is an important step for the extraction of intracellular enzymes, peptides, and other biomolecules. Advances in protein engineering have rapidly accelerated our ability to engineer enzymes, enabled us to perform direct alterations in substrate specificity and enzyme activity. Hence, improvements in cell lysis procedure are crucial for the efficient extraction and purification of such enzymes. Presently, there are many commercial cocktails for cell lysis applications, including B-PER™ (ThermoFisher® SCIENTIFIC), CelLytic™ and BugBuster® (Sigma-Aldrich), SoluLyse™ (Genlantis), as well as mechanical techniques, which exhibit efficient bacterial cell lysis capabilities [8–10]. However, multiple parameters must be considered while devising an effective lysis strategy. Based on the microbial species and the composition of their cell envelopes, diverse lysis methods are employed. For example, mechanical cell disruption (a high-pressure homogenizer) is employed for thick-walled cells like microalgae; enzyme assisted cell lysis is used for plant cells; and detergents are used for animal cell lysis [10, 11]. In addition, gram-positive bacteria with thick cell walls are notably difficult to lyse due to the presence of multiple layers of peptidoglycan polymers cross-linked by teichoic acid [12]. On the contrary, gram-negative cells have an outer membrane while cyanobacterial cells possess a thick exopolysaccharide layer [12]. As a result of this anatomical variability, efficacious and less severe lysis strategies are required to obtain maximum yields of functional proteins. Employing harsh disruptive techniques can overcome the obstacles posed by complex bacterial envelopes, thereby successfully lysing the bacterial cells. However, these techniques can simultaneously reduce the molecular functionality, and ultimately undermining the whole purpose of cell disruption. As it is known that freezing cells lead to decrease in cell viability [7, 13], we hypothesize that storing microbial biomass at ULT can provide the added benefit of improving cell lysis efficiency by avoiding harsh treatments. On the other hand, cultivating fresh cells several time to examine various enzyme parameters can lead to significant deviations in the results owing to experimental variability or manual errors. Hence, it is paramount to understand the impact of longer periods of cell storage on cell lysis efficiency and protein function.

Considering the importance of ULT-storage of microbial biomass in biomolecular studies, we have analyzed the impact of ULT-storage on cell viability of three different microbial species that possess diverse cellular envelope composition and structure. Furthermore, we have demonstrated the correlation of cell viability after ULT storage to protein extraction efficiency.

## Materials and methods

### Chemicals and reagents

Thiazolyl blue tetrazolium bromide (MTT), culture media components, and all the other chemicals were purchased from Sigma-Aldrich (St. Louis, MO, USA). SoluLyse™ was purchased from Genlantis (San Diego, CA). Bradford's reagent was purchased from BioRad. Propidium iodide (PI) was purchased from G-Biosciences, Geno Technology Inc (St. Louis, MO, USA).

### Bacterial strains and cultivation

*Escherichia coli* (hereafter *E. coli*), *Bacillus subtilis* (hereafter *B. subtilis)* were cultivated in LB medium at 37˚C and 250 rpm for 18 hours [14]. *Synechocystis sp*. PCC 6803 (hereafter *Synechocystis* 6803) was grown in BG-11 medium at 30˚C and 250 rpm under 100 μmol/m$^2$/s light intensity for 4 days [15]. These bacterial cells were subjected to PI assay and MTT viability assay. *E. coli* BL21-DE3 transformants expressing mCherry fluorescent protein (BBa_K2033011 plasmid possessing ampicillin resistance; 100 μg/mL) and *E. coli* DH5α transformants expressing eGFP (pZE27GFP–Addgene plasmid #75452 possessing kanamycin resistance; 25 μg/mL), constitutively, were cultivated as above and used for estimating cell lysis efficiency (along with *B. subtilis* and *Synechocystis* 6803) and the impact of ULT storage on protein function.

### ULT-storage

Bacterial cells (*E. coli*, *B. subtilis* and *Synechocystis* 6803) were grown under the respective cultivation conditions. 5 mL of the cells were harvested, and their cell densities were adjusted to the optical density of 0.15 (for 200 μL) and centrifuged at 12,000g for 5 min. One set of cell pellets was incubated with 200 μL of culture medium and another without the medium, followed by incubating in -80˚C freezer for up to 7 days. The PI assay, MTT assay, and fluorescence analyses were performed over 7 days in triplicates. Glycerol stocks (containing 20% sterile glycerol) were stored as the experimental positive control and analyzed along with the experimental samples at the end of their storage.

### PI assay

Bacterial cell densities (for *E. coli*, *B. subtilis* and *Synechocystis* 6803) were adjusted to the Abs$_{600}$ = 0.15. Cells were stored with and without the culture media (200 μL) in -80˚C. Freshly harvested cells and ULT stored samples obtained at different time points were analyzed for their membrane integrity using propidium iodide (PI) assay [16]. 3 μM PI solution was made in nuclease free water. Samples with the medium were thawed and centrifuged at 12000g for 1 min. All the cell pellets were suspended in 200 μL of PI solution and dispensed into 96-well plates. The plates were incubated at 37˚C for 5 min by shaking at 3 mm amplitude. Spectrofluorometer Infinite® 200 by TECAN was used for fluorescence analysis. The fluorescence was obtained at the excitation wavelength of 530 nm and emission wavelength of 610 nm against the 3 μM PI solution which served as a blank control [17, 18]. The data were plotted as fluorescence against the number of days.

### MTT assay

MTT assay was performed following the protocol developed by Wang *et al.* [19]. Cell pellets were mixed with 20 μL of 5 g/L (w/v) MTT solution in water and immediately incubated at 37˚C for exactly 20 min. Dehydrogenase catalyze the reduction of MTT to MTT-formazan-cell

complex, which could be prominently observed as the purple particles in the suspension. The resulting solution was centrifuged at 12,000g for 2 min and the pellet was suspended in 500 μL DMSO followed by vortexing for 5 min yielding a magenta-colored solution. 40 μL of this solution was diluted with 160 μL DMSO and the absorbance was measured at 550 nm. As the optical density of the formazan complex directly corresponds to the number of viable cells, standard curves were generated for all the selected bacteria by using different cell densities ($Abs_{600}$ = 0.01, 0.02, 0.05, 0.10, 0.15, 0.18) that were prepared from cells in their exponential growth phase. The viable cell densities from the MTT assay were deduced by using the calibration curve corresponding to the bacterium (Fig 2A).

## Protein extraction and quantitation

To investigate the effect of long-term ULT storage, bacterial pellets were stored in -80˚C for 7 days, same as previously mentioned. ULT frozen cells were thawed at room temperature (RT) for 10 min. Fresh and/or ULT frozen cell pellets were mixed with 50 μL SoluLyse™ and slowly vortexed for 10 min [20]. 150 μL of distilled water was added to these lysates. One set of lysates was centrifuged at 15,000g for 2 min to obtain soluble protein fraction as the supernatant. Protein concentrations were estimated for both, whole (uncentrifuged) cell lysates and centrifuged cell lysates using Bradford's assay [21] and the lysates of *E. coli* transformants were further used for fluorescence studies. To investigate the effect of short-term ULT storage, *E. coli* cells expressing fluorescent protein were stored in -80˚C for different time durations (10, 30, 60, 120 mins). The proteins were extracted as described before and quantified via Bradford's assay.

Furthermore, comparative effect of short-term (120 mins) and long-term (24 h, 48 h) storage at -20˚C on cell lysis and protein functionality was also performed and has been reported in the (S1 and S2 Figs).

## Fluorescence spectroscopy

*E. coli* transformants expressing the fluorescent proteins mCherry and eGFP were used as a surrogate to study the impact of ULT storage on protein function. mCherry fluorescence was estimated with excitation at 587 nm and emission at 630 nm, whereas eGFP fluorescence was estimated with excitation at 488 nm and emission at 507 nm [22, 23] using spectrofluorometer Infinite® 200 by TECAN. Fluorescence analysis was performed with 200 μL of freshly harvested intact cells as well as cell lysates (whole and centrifuged) in 96-well plates.

## Statistical analysis

Mean values and standard deviations were calculated by Microsoft Excel standard functions. *P*-values used for determining statistical significance of our results were calculated in Microsoft Excel using Student's t-test.

# Results and discussion

## Impact of ULT-storage on bacterial cell viability

ULT storage of harvested microbial biomass is a routine practice in biological laboratories. Biomass is usually stored during transport or long-term experimentation for subsequent processing of the samples. However, its impact on microbial cells is minimally reported. Hence, we sought to perform systematic study to investigate the effect of ULT storage on distinct bacterial platforms in terms of storage duration, cell wall composition, etc. The effect of ULT storage on the chosen microbial platforms was estimated based on cell membrane integrity and

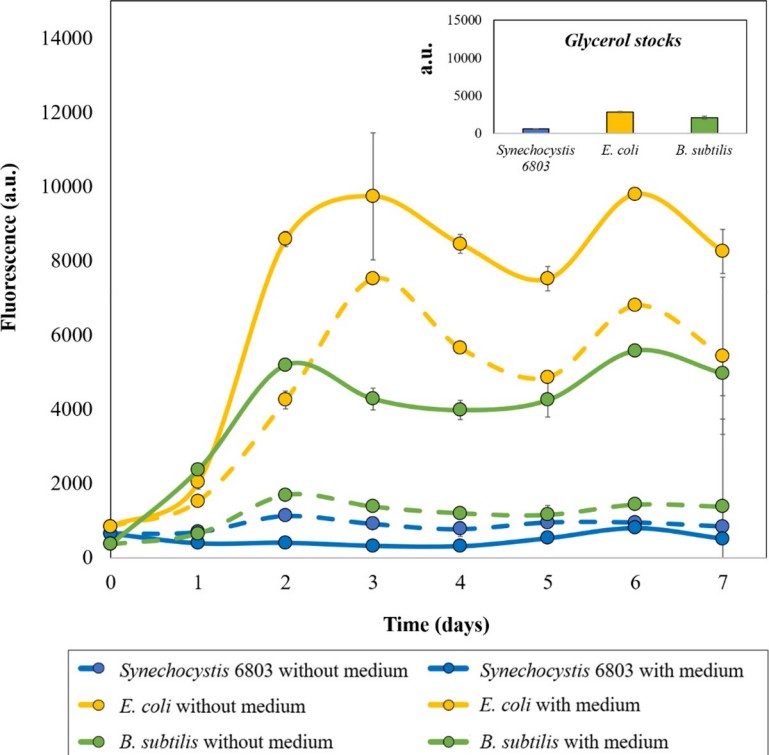

**Fig 1. Effect of ultra-low temperature (ULT) storage on bacterial cell membrane integrity.** The cells stored over a period of 7 days at ULT were subjected to a PI assay. Actively growing bacterial cell membranes pose a barrier for PI entry within the cell, however it can penetrate through membrane compromised or dead bacterial cells and intercalate within their polynucleotides yielding active fluorescence. Our results clearly indicated that with increased storage at ULT, bacterial cell envelopes were damaged, thereby increasing PI entry within the cells and hence increased fluorescence. Highest damage was observed in *E. coli* cells, followed by *B. subtilis* and *Synechocystis* 6803. Also, cells stored with the medium displayed higher fluorescence than the cells stored without the culture medium. Data are shown as mean ± S.D, n = 3.

viability using PI and MTT assays, respectively. The following three bacterial candidates with different cell envelope structure and composition were selected: *E. coli* (Gram-negative); *Bacillus subtilis* (Gram-positive); and *Synechocystis* sp. PCC 6803 (Gram-negative cyanobacterium). Cells were grown under their respective optimal culture conditions (see materials and methods), the biomass was harvested and frozen at -80°C for defined time periods and the corresponding results were correlated to the cell lysis efficiency. Cells were stored with and without culture medium to study the effect of frozen aqueous medium on cellular integrity.

The cells stored at ULT were subjected to a PI assay. PI is a fluorescent intercalating dye which is excluded by viable cells and is widely used for distinguishing live bacteria from the dead ones. Therefore, the PI assay could be used as an indicator of cell envelope integrity, especially the cell membrane [24]. Our results indicated that within 2 days of storage, fluorescence was found to increase significantly for *E. coli* cells, followed by *B. subtilis* (Fig 1). *Synechocystis* 6803 pellets did not display any significant change in the fluorescence over the storage duration. Moreover, cells stored in the growth medium yielded higher fluorescence readings as compared to cells stored without the medium for both *E. coli* and *B. subtilis* (Fig 1). This distinctive response could be attributed to the additional effect of extracellular media crystallization weakening the cell envelope [25]. The assay confirmed that the bacterial cell membranes were compromised with increased storage under ULT. However, PI can pass through the cell

envelopes of actively growing cells in miniscule amounts, but is actively exported, unlike cells with weakened cell walls or dead cells [26]. It has also been demonstrated that PI might in some cases provide false dead signals correlating to high membrane potential due to cell physiology and not the membrane damage [24]. Apart from this, unbound PI has been found to possess strong background fluorescence of 400–500 [27]. These background signals could not be prevented in fluorescence readouts but were instead negated by subtracting the signal obtained from the blank control [27]. Considering these limitations of PI assay, we further verified the effect of ULT using MTT viability assay, according to the protocol developed by Wang *et al.* [19].

As a next step, the biomass of all three bacterial candidates stored at ULT were subjected to an MTT assay. In the MTT assay, viable cell density was primarily estimated from the standard curves (Abs$_{550}$ v/s Abs$_{600}$) of the corresponding species prepared from actively growing cells (Fig 2A). Relatively high correlation coefficient (close to 1) indicated the reliability of using the standard curves for assessing cell viability.

Relative (%) cell viability based on MTT assay revealed that *E. coli* cells significantly lost their viability when incubated with the culture medium as compared to the cells stored as pellets (i.e., without the culture medium) within 24 hours of storage at -80˚C (Fig 2B). This trend was not observed with the other two bacterial candidates and hence this can be attributed to the gram-negative cell wall structure of *E. coli* and the rupture caused by extracellular shear forces originating from ice crystals [25]. On the other hand, *E. coli* cell pellets retained significant viability for 3 days and the cell viability dropped over the next 4 days of storage. In

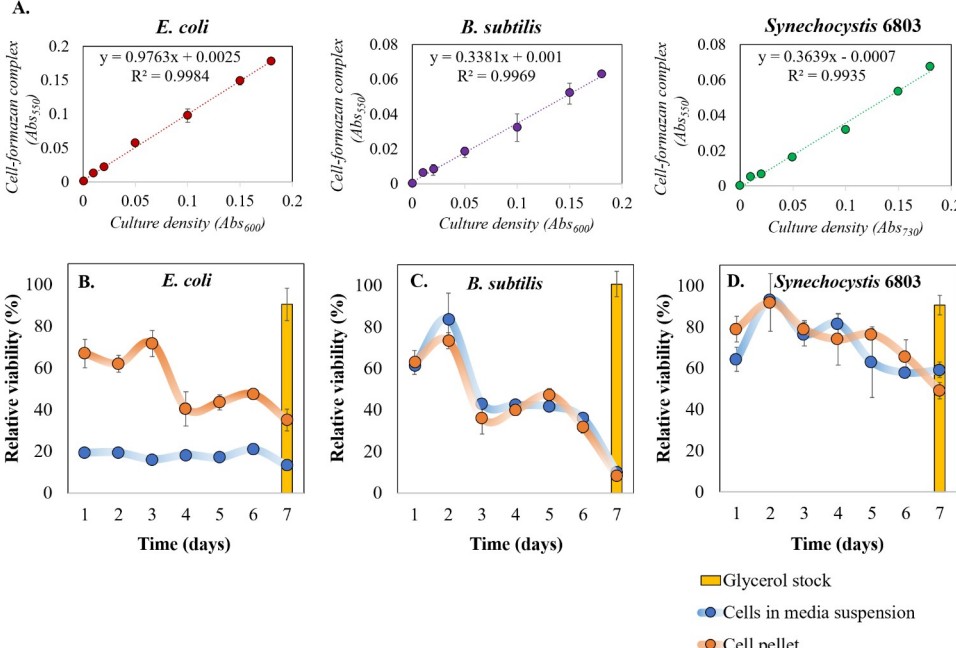

**Fig 2. Impact of storing cells at ultra-low temperature on cell viability.** The cells stored at ULT over a period of 7 days were subjected to MTT assay. MTT assay is based on a simple principle of active dehydrogenase present in viable cells, whereby such active cells can convert MTT to MTT-formazan-cell complex (solubilized in DMSO), which has a $\lambda_{max}$ of 550 nm. This establishes direct correlation between cell viability and absorbance at 550 nm. We hereby present MTT assay standard curves for bacteria with different cell densities (measured by Abs$_{600}$ or Abs$_{730}$) (A), and relative (%) viability plots (with respect to that of freshly harvested cells) for *E. coli* (B), *B. subtilis* (C), *Synechocystis* 6803 (D) with (blue) and without (orange) the culture media. Relative viabilities were also compared with their glycerol stocks (bars in the plots) which had almost 100% viability. Data are shown as mean ± S.D, n = 3.

correlation with the PI assay results, it can be stated that although the *E. coli* membrane integrity was found to be significantly compromised on day 2, the cell viability was still retained at ~60% based on MTT assay results (Figs 1 and 2B). More importantly, both of these assays clearly indicated the negative impact of extracellular aqueous medium on the membrane integrity and cell viability (Figs 1 and 2B).

Like *E. coli*, *B. subtilis* cells exhibited effective viability for the first two days (60–80%), with and without the culture medium. Following the initial two days, % viability dropped to almost ~40% on the 3rd day and to ~10% by the 7th day (Fig 2B and 2C). This differential response to the presence of aqueous media by *B. subtilis* could be correlated to the presence of a relatively thick peptidoglycan layer in its cell wall (Figs 1 and 2C) [12]. The gradual loss of viability would be the result of cryodamage caused by intra- and extracellular crystallization during the storage and can also be attributed to osmotic pressure changes due to crystallization [28].

On the other hand, *Synechocystis* 6803 cells retained significant membrane integrity and cell viability over the entire duration of the experiment irrespective of storage with or without the medium (Figs 1 and 2D). In an earlier study, Lin *et al.* [29] had demonstrated that *Synechocystis* 6803 retains ~41.8% of photosynthetic quantum yield, even after freezing, which correlates with our current observation. Overall, these findings support the fact that storing cells at -80˚C negatively impacts cell viability to varying degrees depending on the structure and composition of the cell envelope. In past studies, it has been observed that gram negative organisms are more prone to cryodamage than gram positive ones [7, 30]. However, the current study is not in complete agreement with this report: cyanobacteria, despite being gram negative, displayed highest survival and membrane rigidity owing to their distinct cellular architecture [30, 31].

## Impact of ULT-storage on protein extraction

Reduced membrane integrity and decreased cell viability as observed in Figs 1 and 2 during storage at -80˚C indicated that weakened cell envelopes may aid cell lysis and improve protein extraction efficiency. To verify this hypothesis, as a next step we studied the impact of storing cells at -80˚C on protein extraction. We used SoluLyse™ for protein extraction and the protein concentrations in the resulting cell lysates were estimated using Bradford's assay (Fig 3A). Our first observation was that whole cell lysate possessing cell debris displayed higher protein concentration owing to the presence of insoluble protein fractions in comparison to the supernatant that contained only the soluble protein fraction (Fig 3B & 3C). In addition, an expected trend was observed between protein concentration (which is an indirect measure of cell lysis efficiency) and the cell envelope composition of bacterial cells. *E. coli* cells with relatively weaker cell envelope system displayed higher degree of cell lysis (i.e., higher protein concentration). This was followed by *B. subtilis* and *Synechocystis* 6803; both these cells displayed only a miniscule cell breakage (Fig 3C). As previously discussed, *B. subtilis* possesses a thick peptidoglycan layer and is relatively difficult to break-open, which is precisely reflected from the protein concentration in the cell lysate. *Synechocystis* 6803, on the other hand, has a unique extracellular matrix (S-layer) and multi-membrane (thylakoid) system within the cell that not only confers extended viability at -80˚C, but is recalcitrant to the disruption steps employed to extract the intracellular proteins [32]. Hence, it can be corroborated that along with external physical stress and the presence of an outer membrane, intracellular organelle architecture, like thylakoid membranes may influence cell morphology, cell survival, cell lysis and therefore, protein extraction efficiency [33]. In addition, by comparing these results to Figs 1 and 2, it can also be concluded that

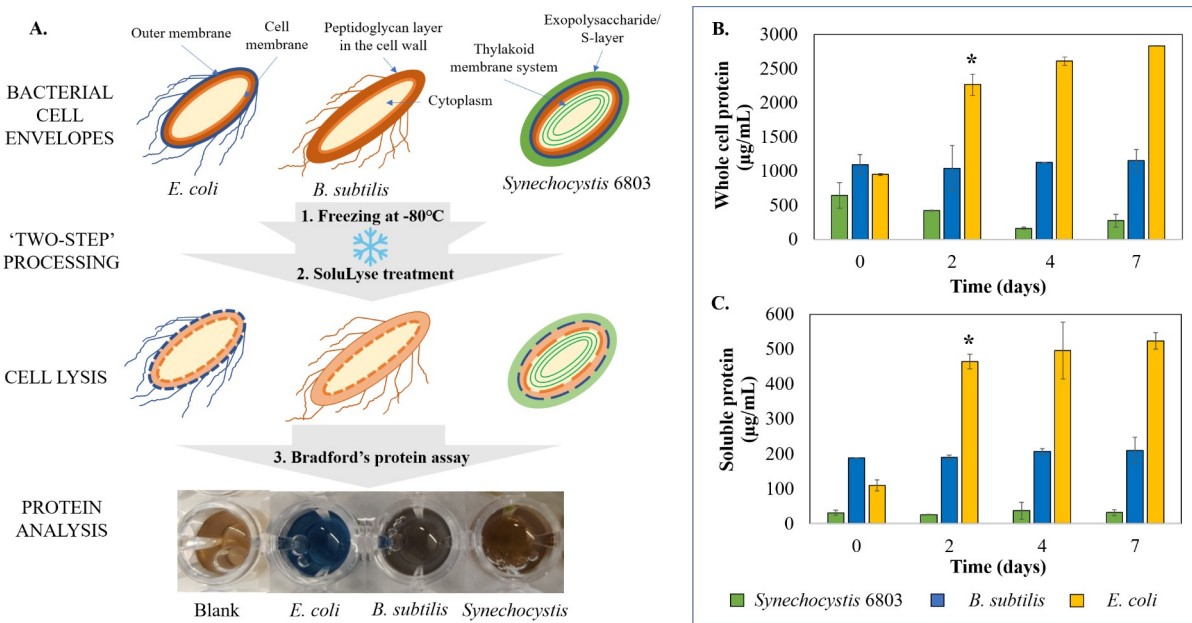

**Fig 3. Impact of ULT storage on protein extraction efficiency.** A) Schematic representation of bacterial cell structures used in this study and the steps followed for protein extraction and analysis. Comparative protein concentrations of whole (B) and soluble fraction of (C) cell lysates from *E. coli* (yellow), *Synechocystis* 6803 (green) and *B. subtilis* (blue) over 7 days of ULT-storage. *E. coli* significantly displayed highest protein yields. The experiments were performed in biological triplicates. Statistical significance was determined using t-test (p<0.05). Data represented as Mean ± SD, n = 3.

there is a direct correlation between bacterial membrane integrity, cell viability during -80˚C storage and cell lysis efficiency, more significantly for *E. coli* cells.

## Impact of ULT-storage on protein function

Thus far, we established that storing cells in a frozen state negatively impacted cell viability enabling improved protein extraction. Although improving protein extraction is critical for several fields, it is equally important to make sure that the biomolecules are functional during the low temperature storage and post-extraction. As *E. coli* is the major microbial host used for protein expression studies, this part of the study was restricted only to *E. coli* strains. *E. coli* cells engineered to constitutively express fluorescent proteins mCherry and eGFP were used to estimate the effect of -80˚C storage on protein function. Total fluorescence from the whole cell lysates and from their soluble fractions (supernatant) were estimated. Relative fluorescence was calculated for the whole cell lysate, as a percentage of protein extracted from freshly harvested cells, to analyze the effect of ULT-storage on the protein function. Fluorescence estimation with the whole cell lysates indicated that the relative (%) fluorescence was maintained at an average of 100 ± 3% for both the proteins (Fig 4; dots) across all time points. This observation confirmed that the protein function was not affected during the entire storage period of the microbial biomass. In addition, fluorescence intensity (a.u.) of the soluble protein fraction (Fig 4; bars) precisely correlated with the protein concentration trend observed in Fig 3C showing proper extraction of intracellular soluble proteins. Therefore, storing cells at ULT can be harnessed for long term protein storage as the function of the protein within the unruptured cell is intact. Plus, it can provide the added benefit of obtaining improved protein yield from the cells stored at ULT.

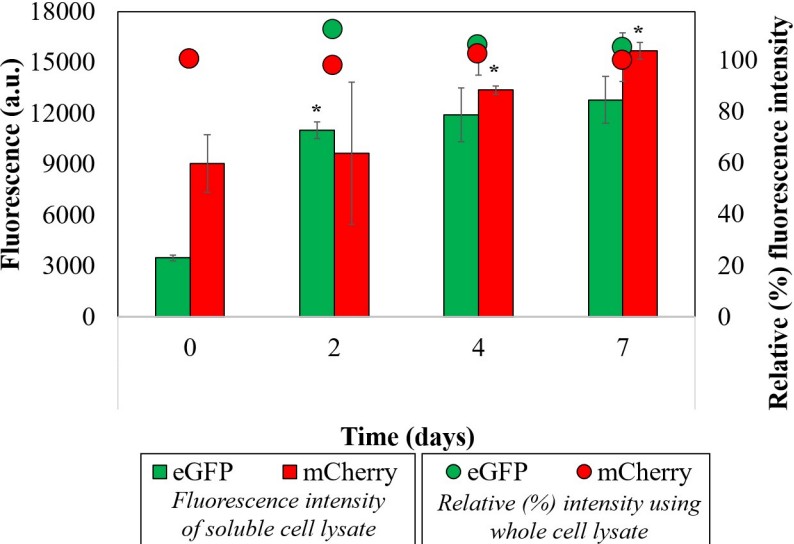

**Fig 4. Fluorescence analysis of the extracted proteins.** Fluorescence intensity of soluble protein lysates (bars) indicates increase in protein extraction efficiency with the duration of ULT-storage. Quantification of functional proteins was performed using whole cell lysates to determine relative fluorescence intensity (dots) of protein samples. Relative fluorescence intensity was calculated as a ratio of proteins present in whole cell lysates from any day to that of day 0. The experiments were performed in biological triplicates. Statistical significance was determined using t-test (5% level of significance). Data represented as Mean ± SD, n = 3.

## ULT storage as a practical step to improve protein extraction

The findings discussed in the above sections clearly convey that *E. coli* cells exhibited improved cell lysis and protein yield just within 2 days of ULT storage (Fig 3C). As ULT storage is a simple step, this could very well be utilized to improve protein (or other biomolecules) extraction efficiency. However, storing cells for 2 days with the goal to improve protein extraction from cells is not time efficient. To understand if a shorter more practical storage time exists, we analyzed the effect of short-term ULT storage on protein extraction efficiency using *E. coli* transformants expressing fluorescent proteins. Interestingly, storing *E. coli* strains expressing mCherry just for 10 and 120 minutes resulted in 2.7-fold (from 93 μg/ml to 254 μg/ml) and 4-fold (93 μg/ml to 380 μg/ml) improvement in the protein extraction efficiency, respectively (Fig 5). Likewise, storing *E. coli* strains expressing eGFP for 120 minutes resulted in approximately 2-fold (310 μg/ml to 602 μg/ml) increase in protein extraction (Fig 5). In addition, as expected from Fig 4, fluorescence analysis indicated the protein function to be intact (Fig 5).

-20˚C storage is another common storage temperature employed in biological laboratories and it has been proven to have detrimental effect on bacterial cell viability [7]. Therefore, a short comparative study was conducted between -80˚C and -20˚C storage of microbial cells. The results indicated that -20˚C storage yielded relatively less amount of protein from *E. coli* cells during both short-term (< 120 mins) as well as long-term (> 1 day) storage (S1 Fig). However, the extracted protein concentration is substantially higher than that from the freshly harvested cells, without significantly impacting the protein functionality (S2 Fig). Therefore, either of these low temperatures can be practically implemented for the storage of bacterial cell and to improve protein extraction. Overall, these findings suggest that a more practical ULT storage time of 10 to 120 minutes can be adapted by researchers to improve protein extraction efficiency by several folds.

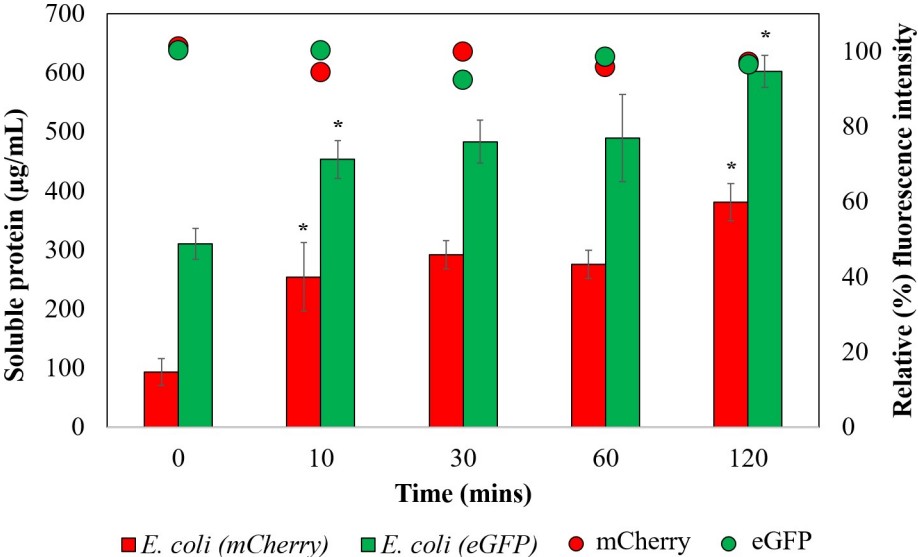

**Fig 5. Impact of short-term ULT storage on protein extraction efficiency and functionality.** Comparative protein concentrations of the soluble fraction of cell lysates from *E. coli* strains expressing mCherry (red) and eGFP (green) over 0, 10, 30, 60, 120 mins of ULT-storage. Both the *E. coli* strains displayed significant improvement in protein extraction after ULT-storage. Fluorescence analyses further confirmed sustained protein functionality. The experiments were performed in biological triplicates. Statistical significance was determined using t-test (5% level of significance). Data represented as Mean ± SD, n = 3.

## Conclusions

In this study, we have successfully demonstrated the effect of -80°C storage on cell viability of different bacterial systems are widely used as platform strains for different metabolic and protein engineering applications. The strains differ from each other mainly in their cell envelope structure and composition. Through detailed analysis, we have established a possible correlation between the cell envelope structure and composition to cell lysis following ULT-storage. Advantageously, the study revealed that ULT storage of microbial cells can be employed for long term storage (beyond the 7-day period explored in this study) of functional proteins intracellularly, thereby reducing errors that would occur due to experimental variations upon repetition. On the other hand, this principle can well be applied for improving functional protein extraction efficiency by storing microbial biomass at ULT for a short time and to use them for *in vitro* assays, his-tag purification, etc. Upon further studies, the procedure can be effectively extended to tough cyanobacterial or gram-positive cells by supplementing lysis mix with specific hydrolyzing enzymes like cellulase, lysozyme, etc., followed by ULT storage to improve the protein extraction efficiency. Apart from ULT storage time it is known that multiple other parameters have collaborative effect on bacterial cell lysis including suspending medium, the rate of freezing and thawing etc. [34, 35]. Through further research, these parameters can be optimized to improve the extraction efficiency of biomolecules further and also to lyse bacterial cells with complex cell envelope.

Researchers often utilize ULT storage for storing the harvested microbial biomass during transport or for subsequent protein (or other biomolecular) extraction during the long-term experiment; but ignore its impact on the results of their following experimentation. As this investigation clearly establishes ULT storage time to impact protein extraction efficiency, we envision that this work will serve as a foundation upon which researchers can strategize their future experimental studies.

## Supporting information

**S1 Fig. Comparative analysis of -80˚C and -20˚C storage of *E. coli* biomass.** Comparative analysis of -80˚C and -20˚C storage of *E. coli* biomass for (A) short-term, 120 mins; and (B) long-term, 24 h and 48 h revealed that the lysis efficiency is relatively higher with -80˚C storage than -20˚C. (DOCX)

**S2 Fig. Effect of -20˚C storage of *E. coli* biomass for short-term (120 mins) and long-term (24 h and 48 h).** Effect of -20˚C storage of *E. coli* biomass for short-term (120 mins) and long-term (24 h and 48 h) indicated that *E. coli* (mCherry) strains exhibited 4.6-folds improvement in the concentration of extracted protein after 48h, whereas *E. coli* (eGFP) showed 1.8-folds increase in the extracted protein concentration over that from the freshly harvested biomass. (DOCX)

**S1 Data.**
(XLSX)

## Acknowledgments

We thank Dr. David Nielsen, Zachary Dookeran and Cody Kamoku (Arizona State University, Tempe, AZ, USA) for providing us with the *E. coli* strains constitutively expressing *eGFP* and *mCherry*. We acknowledge Mark Nguyen (Arizona State University, Tempe) for his time and efforts during protein extraction and quantitation. Sandia National Laboratories is a multimission laboratory managed and operated by National Technology & Engineering Solutions of Sandia, LLC, a wholly owned subsidiary of Honeywell International Inc., for the U.S. Department of Energy's National Nuclear Security Administration under contract DE-NA0003525.

## Author Contributions

**Conceptualization:** Aditya Sarnaik, Ryan Davis, Arul M. Varman.

**Data curation:** Aditya Sarnaik, Arul M. Varman.

**Formal analysis:** Aditya Sarnaik, Dylan Smith.

**Funding acquisition:** Ryan Davis, Arul M. Varman.

**Investigation:** Aditya Sarnaik, Apurv Mhatre, Muhammad Faisal, Dylan Smith.

**Methodology:** Aditya Sarnaik, Apurv Mhatre, Muhammad Faisal, Dylan Smith.

**Project administration:** Aditya Sarnaik, Arul M. Varman.

**Resources:** Aditya Sarnaik, Arul M. Varman.

**Supervision:** Arul M. Varman.

**Validation:** Aditya Sarnaik, Arul M. Varman.

**Visualization:** Aditya Sarnaik.

**Writing – original draft:** Aditya Sarnaik.

**Writing – review & editing:** Aditya Sarnaik, Ryan Davis, Arul M. Varman.

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
