## [Decision Letter · Decision Letter 0]

16 Dec 2020

PONE-D-20-35156

Novel perspective to a conventional technique: Impact of ultra-low temperature on bacterial viability and protein extraction

PLOS ONE

Dear Dr. Varman,

Thank you for submitting your manuscript to PLOS ONE. After careful consideration, we feel that it has merit but does not fully meet PLOS ONE’s publication criteria as it currently stands. Therefore, we invite you to submit a revised version of the manuscript that addresses the points raised during the review process.

Please submit your revised manuscript within 3 months. If you will need more time than this to complete your revisions, please reply to this message or contact the journal office at plosone@plos.org. Please include the following items when submitting your revised manuscript:

We look forward to receiving your revised manuscript.

Kind regards,

Mehmet A Orman

Academic Editor

PLOS ONE

Journal Requirements:

Reviewers' comments:

Reviewer's Responses to Questions

**Comments to the Author**

1. Is the manuscript technically sound, and do the data support the conclusions?

Reviewer #1: Yes

Reviewer #2: Partly

2. Has the statistical analysis been performed appropriately and rigorously? 

Reviewer #1: Yes

Reviewer #2: No

3. Have the authors made all data underlying the findings in their manuscript fully available?

Reviewer #1: Yes

Reviewer #2: Yes

4. Is the manuscript presented in an intelligible fashion and written in standard English?

Reviewer #1: Yes

Reviewer #2: Yes

5. Review Comments to the Author

Reviewer #1: Description: In this manuscript, Sarnaik et al. investigated the effect of ultra-low temperature storage on the membrane-integrity/cell-viability as well as the correlation with protein extraction of three distinct prokaryotic platforms: E. coli, B. subtilis, and Synechocystis sp. PCC 6803. They have provided evidence that have showed depending on the storage time of these prokaryotic cells in -80 °C, the cell viability is altered, that can affect the protein extraction efficiency. The work is very comprehensive and the manuscript is well written. However, we have certain concerns that can be addressed:

Comments:

1. In lines 245-248 in the conclusion, this manuscript suggested that ULT storage time can impact protein extraction efficiency that can be used for experimental studies. There are several questions that needs to be addressed regarding this statement:

a. Generally, using -80 °C preserved cells, a primary culture is set overnight, followed by a secondary culture and induction. By the time the cells are harvested, they have already passed through several generations, so does the statement that ‘ULT storage time can impact protein extraction efficiency’ hold true?

b. Are the authors suggesting that the harvested cells can be stored in ULT storage for some time (0-7days) before extraction? If that be the case, then it is worth mentioning that several labs across the globe do not allow access to -80 °C for freeze-thaw cycles. If someone wants to use it quickly (say within 2 days), but the lab facility doesn’t allow easy access to ultra-storage facility, then this method can basically delay the experiment. What are the author’s suggestion to use -20 °C in this case? Will storage in -20 °C for quick and easy access of cells have the same effect?

c. Even if someone uses this method of freeze-thawing of cells harvested in bulk for protein extraction, how does this method differ from other conventional methods of protein extraction?

d. Does the glycerol percentage used during the storage of the cells affect the membrane stability? The current study has been done with 20% glycerol. However, for labs that use 25%-30% glycerol for ultra-low temperature storage, will the same hold true?

2. Line 35: “Interestingly, E. coli exhibited concomitant increase in cell lysis efficiency resulting in a 5-folds increase in the extracted protein titer just after 2 days of ULT storage, followed by Bacillus and Synechocystis.” The increase in the extracted titre is not expanded in the result section. Mentioning the value with respect to wild type will help to understand the proper quantification of the titre.

3. Grammatical errors need to be corrected in

line 63: Hence, improvements in cell lysis procedures has become crucial during extraction and purification of such enzymes.

4. Line 72-73: Authors have mentioned about the difficulty of bacterial cell lysis with a comparison with microalgae, plant and animal cell. Authors can elaborate on possible causes of this complexity in bacterial cells, emphasising on the difference with plant and animal cells.

5. Line 101: In MTT assay section, authors should follow one tense (Past/Present) for better readability. Overall, throughout the “materials and methods section”, one convention must be followed.

6. Line 113: In ULT storage section, authors have not mentioned how much volume of the culture was used to prepare the cell pellet. Does this method work irrespective of cell volume? Please clarify

7. Line 204: Authors have mentioned that intracellular cell organelles play critical role in lysis of the cell. Explanation of this statement will clarify which organelle usually controls the cell lysis and how it regulates the process.

8. Authors have conducted experiments for a time span of seven days. In discussion section authors can highlight their views on cellular envelope integrity with longer storage time of more than 7 days.

Reviewer #2: Through their work, the authors have shown the impacts of ULT-storage in bacterial cells using three different strains with varying cell wall composition. They conclude that, depending on the bacterial strains, ULT-storage can affect cell viability and lysis. Although the study verifies and validates the impacts of low temperate storage, it doesn’t provide any methods for improving the standard ULT-storage protocol. The conclusions drawn in the manuscript has been noted in previous studies.

The authors state that gram-negative bacteria are more susceptible to ULT-storage however, testing another gram-negative bacterium would make the claim stronger.

The manuscript also demonstrates that decreased cell viability during ULT-storage results in more efficient cell lysis. This can be an effect of the fraction of cells with a compromised membrane in the culture. In addition to MTT assay, performing propidium iodide (PI) staining in the cell culture would provide a better idea on quantifying the fraction of cells with such compromised membranes.

For the comparison of graphs in figures 2 and 3, the authors have not mentioned what statistical tests were conducted and what the threshold for significance was set to be.

For the fluorescence protein procedure, the authors have used E. coli transformants expressing mCherry and eGFP. The manuscript does not provide the experimental details on generating these transformants.

Finally, although the authors have pointed out various reasons that could have resulted in their observation, they have not been experimentally tested.

6. PLOS authors have the option to publish the peer review history of their article (what does this mean?). If published, this will include your full peer review and any attached files.

Reviewer #1: No

Reviewer #2: No

---

## [Author Response · Author response to Decision Letter 0]

3 Apr 2021

Response to the reviewer

Reviewer #1: 

Description: In this manuscript, Sarnaik et al. investigated the effect of ultra-low temperature storage on the membrane-integrity/cell-viability as well as the correlation with protein extraction of three distinct prokaryotic platforms: E. coli, B. subtilis, and Synechocystis sp. PCC 6803. They have provided evidence that have showed depending on the storage time of these prokaryotic cells in -80 °C, the cell viability is altered, that can affect the protein extraction efficiency. The work is very comprehensive, and the manuscript is well written. However, we have certain concerns that can be addressed:

Comments:

1. In lines 245-248 in the conclusion, this manuscript suggested that ULT storage time can impact protein extraction efficiency that can be used for experimental studies. There are several questions that needs to be addressed regarding this statement:

a. Generally, using -80 °C preserved cells, a primary culture is set overnight, followed by a secondary culture and induction. 

We thank the reviewer for bringing this point to our notice. We would like to mention here that the current manuscript entails the direct effect of ULT on ‘stored bacterial biomass’ and not ‘preserved cultures’ as stocks, which will be inoculated for growing the seed cultures. As the reviewer pointed out, ULT preservation of microbes and the effect ULT on protein extraction from microbial biomass are two different aspects to be considered. The misconception could be because of the miscommunication from our side. 

By the time the cells are harvested, they have already passed through several generations, so does the statement that ‘ULT storage time can impact protein extraction efficiency’ hold true?

We consider this to be a miscommunication from our side. The sentence should basically convey the effect of ULT storage and not preservation of microbial biomass. Usually, researchers perform one-time collection of biomass under different environmental conditions, time points etc., and store them at ULT before processing them together for the extraction of various intracellular molecules, especially proteins and enzymes. Additionally, while shipping out the samples in cold storage or during long experimental plans/ vacations, we even leave the biomass samples in ULT, for processing them later. Considering this significant time gap between the biomass harvesting and protein extraction, our findings suggest that researchers must take into account the effect of ULT storage on cell viability, protein extraction and function. In this respect, we had made the prior statement. However, now we have modified it as below to avoid this miscommunication.

Abstract:

“Ultra-low temperature (ULT) storage of microbial biomass is routinely practiced in biological laboratories.” Line 25-26

Introduction:

“Ultra-low temperature (ULT) storage of these recombinant cells and harvested biomass is a routine exercise in biological laboratories before subsequent sample processing; but its impact on the cells and proteins of interest has been minimally reported.” Line 57-60

Conclusion:

“Researchers often utilize ULT storage for storing the harvested microbial biomass during transport or for subsequent protein (or other biomolecules) extraction during the long-term experiment; but ignore its impact on the results of their subsequent experimentation.” Line 386-388

b. Are the authors suggesting that the harvested cells can be stored in ULT storage for some time (0-7days) before extraction? 

We thank the reviewer for this question. Yes, our aim is to convey that extended ULT storage time would provide a secondary benefit of increasing protein extraction efficiency. Also, it is to be noted that ULT storage is independent of the method that would be used later for biomolecular extraction. Therefore, our manuscript suggests ULT storage as a ‘processing-independent step’ that has dual advantages - improving the extraction efficiency and storing the protein (inside cells) without altering its function. The primary motivation however is to make the researchers aware of the impact of ULT storage on cell viability, protein extraction and protein function, so they can plan their experiments accordingly to obtain consistent results.

If that be the case, then it is worth mentioning that several labs across the globe do not allow access to -80 °C for freeze-thaw cycles. If someone wants to use it quickly (say within 2 days), but the lab facility doesn’t allow easy access to ultra-storage facility, then this method can basically delay the experiment. What are the author’s suggestion to use -20 °C in this case? Will storage in -20 °C for quick and easy access of cells have the same effect?

We thank the reviewer for the excellent suggestion. Based on this question we incorporated two studies in the revised manuscript: a) comparison between the effect of -80℃ vs -20℃ storage on protein extraction and functionality, b) effect of short-term ULT storage on protein extraction. Results of both the studies have been incorporated in the manuscript as follows. 

“-20℃ storage is another common storage temperature employed in biological laboratories and it has been proven to have detrimental effect on bacterial cell viability (7). Therefore, a short comparative study was conducted between -80℃ and -20℃ storage of microbial cells. The results indicated that -20℃ storage yielded relatively less amount of protein from E. coli cells during both short-term (< 120 mins) as well as long-term (> 1 day) storage (Figs. SI-1A and 1B). However, the extracted protein concentration is substantially higher than that from the freshly harvested cells, without significantly impacting the protein functionality (Fig SI-2). Therefore, either of these low temperatures can be practically implemented for the storage of bacterial cell and to improve protein extraction. Overall, these findings suggest that a more practical ULT storage time of 10 to 120 minutes can be adapted by researchers to improve protein extraction efficiency by several folds.” Line 356-365

Here are the plots for reviewer’s kind perusal: 

 Short-term storage 

-80℃ v/s -20℃ storage

c. Even if someone uses this method of freeze-thawing of cells harvested in bulk for protein extraction, how does this method differ from other conventional methods of protein extraction? 

We thank the reviewer for getting this clarified. The primary motivation behind this manuscript is to fill in the knowledge-gap between the extended ULT storage of harvested biomass and its impact on the efficiency of extraction. The protein extraction protocol will still remain the method of one’s choice/preference. Our secondary suggestion is if researchers have time in their hand, they can include an extended ULT storage to any method of their choice to improve protein extraction. 

d. Does the glycerol percentage used during the storage of the cells affect the membrane stability? The current study has been done with 20% glycerol. However, for labs that use 25%-30% glycerol for ultra-low temperature storage, will the same hold true?

20% glycerol storage was used as one of the positive controls for the experiment and it could have been very well 25% or 30% glycerol as well. Therefore, as they are only controls, our findings would still hold true. 

To avoid confusion with the glycerol stocks, we have modified the sentence as follows:

“The PI assay, MTT assay and protein analyses were performed over 7 days in triplicates including freshly pelleted cells which were used as the reference to estimate the effect of cell storage. Glycerol stocks (containing 20% sterile glycerol) were stored as the experimental positive control and analyzed along with the experimental samples at the end of 7 days of storage.” Line 122-125

2. Line 35: “Interestingly, E. coli exhibited concomitant increase in cell lysis efficiency resulting in a 5-folds increase in the extracted protein titer just after 2 days of ULT storage, followed by Bacillus and Synechocystis.” The increase in the extracted titre is not expanded in the result section. Mentioning the value with respect to wild type will help to understand the proper quantification of the titre.

We thank the reviewer for the suggestion. We have modified the statement to include the protein concentration values.

Abstract:

“Interestingly, E. coli exhibited concomitant increase in cell lysis efficiency resulting in a 4.5-fold increase (from 109 µg/ml of protein on day 0 to 464 µg/ml of protein on day 2) in the extracted protein titer following ULT storage”. Line 37-40

Based on new experiments following statement has been included in the modified abstract:

“These results established the plausibility of using ULT storage to improve protein extraction efficiency. Towards this, the impact of shorter ULT storage time was investigated to make the strategy more time efficient to be adopted into protocols. Interestingly, E. coli transformants expressing mCherry yielded 2.7-folds increase (93 µg/mL to 254 µg/mL) after 10 mins, while 4-folds increase (380 µg/mL) after 120 mins of ULT storage in the extracted soluble protein.” Line 41-46

3. Grammatical errors need to be corrected in

line 63: Hence, improvements in cell lysis procedures has become crucial during extraction and purification of such enzymes.

We have corrected the suggested error. Also, we checked through the entire manuscript for any other corrections to be made and corrected them.

“Hence, improvements in cell lysis procedure are crucial for the efficient extraction and purification of such enzymes.” Line 71-72

4. Line 72-73: Authors have mentioned about the difficulty of bacterial cell lysis with a comparison with microalgae, plant and animal cell. Authors can elaborate on possible causes of this complexity in bacterial cells, emphasizing on the difference with plant and animal cells.

We appreciate the suggestion by the reviewer. Basically, we want to bring to the reader’s notice that like plant or fungal or microalgal cell walls, bacterial cell walls are equally difficult to break open. However, the complexity rises when the cell wall composition changes significantly. 

Therefore, considering your suggestion and our point of view, we have modified the sentence as;

“In addition, gram-positive bacteria with thick cell walls are notably difficult to lyse due to the presence of multiple layers of peptidoglycan polymers cross-linked by teichoic acid (12). On the contrary, gram-negative cells have an outer membrane while cyanobacterial cells possess a thick exopolysaccharide layer (12). As a result of these physiological variability efficacious and less severe lysis strategies are required to obtain maximum yields of functional proteins.” Line 79-84

5. Line 101: In MTT assay section, authors should follow one tense (Past/Present) for better readability. Overall, throughout the “materials and methods section”, one convention must be followed.

We thank the reviewer for the suggestion. The suggestion has been duly considered in the revised manuscript and changes have been made with respect to the tense, maintaining uniformity of the text. Please take a look at the Materials and Methods section. Line 99-179

6. Line 113: In ULT storage section, authors have not mentioned how much volume of the culture was used to prepare the cell pellet. Does this method work irrespective of cell volume? Please clarify

We thank the reviewer for identifying this omission. The modified sentence from the manuscript is provided below for your convenience.

“5ml of the cells were harvested and their cell densities were adjusted to the optical density of 0.15 (for 200µl) and centrifuged at 12,000g for 5 min.” Line 119-120

The 2nd part is an excellent question. If the cell volume is changed, it will change the freeze-thawing rate. As our study provides a direct correlation between cell death and protein extraction efficiency, our findings will hold true irrespective of the cell volume. However, the improvement percentage would differ from the study we had conducted. 

7. Line 204: Authors have mentioned that intracellular cell organelles play critical role in lysis of the cell. Explanation of this statement will clarify which organelle usually controls the cell lysis and how it regulates the process.

We thank the reviewer for the suggestion. We have discussed about intracellular organelles present in cyanobacteria, to offer an explanation for the reduced cell lysis that was observed. To make this point clear, we have mentioned about thylakoid membrane in the text for cyanobacteria. Please find below the edited text from the manuscript that addresses your suggestion

“Synechocystis 6803, on the other hand, has a unique extracellular matrix (S-layer) and multi-membrane (thylakoid) system within the cell that not only confers extended viability at -80℃, but is recalcitrant to disruption to extract the intracellular proteins (32). Hence, it can be corroborated that along with external physical stress and the presence of an outer membrane, intracellular organelle architecture, like thylakoid membranes may influence cell morphology, cell survival, cell lysis and therefore, protein extraction efficiency.” Line 289-294

8. Authors have conducted experiments for a time span of seven days. In discussion section authors can highlight their views on cellular envelope integrity with longer storage time of more than 7 days.

This is an excellent suggestion. Please find the modified sentence below.

“Advantageously, the study revealed that ULT storage of microbial cells can be employed for long term storage (beyond the 7-day period explored in this study) of functional proteins intracellularly, thereby reducing errors that would occur due to experimental variations upon repetition.” Line 373-376

Reviewer #2: 

Through their work, the authors have shown the impacts of ULT-storage in bacterial cells using three different strains with varying cell wall composition. They conclude that, depending on the bacterial strains, ULT-storage can affect cell viability and lysis. Although the study verifies and validates the impacts of low temperate storage, it doesn’t provide any methods for improving the standard ULT-storage protocol. The conclusions drawn in the manuscript has been noted in previous studies

We thank the reviewer for bringing this point to our notice. Based on your suggestions we have revised our manuscript and addressed your comments in the revised version. 

The authors state that gram-negative bacteria are more susceptible to ULT-storage however, testing another gram-negative bacterium would make the claim stronger.

We appreciate the suggestion provided by the reviewer and agree to it. Indeed, in this work we have explored two gram-negative organisms E. coli and Synechocystis (cyanobacteria). However, both gram-negative yielded varying results. Overall, multiple variables play a role in cell lysis and we have included the following sentences along with appropriate citations to capture the complete analysis. 

“This trend was not observed with the other two bacterial candidates and hence this can be attributed to the gram-negative cell wall structure of E. coli…” Line 246-248

“Overall, these findings support the fact that storing cells at -80℃ negatively impacts cell viability to varying degrees depending on the structure and composition of the cell envelope. In past studies, it has been observed that gram negative organisms are more prone to cryodamage than gram positive ones (7, 30). However, the current study is not in complete agreement with this report: cyanobacteria, despite being gram negative, displayed highest survival and membrane rigidity owing to their distinct cellular architecture (30, 31).” Line 267-272

“Apart from ULT storage time it is known that multiple other parameters have collaborative effect on bacterial cell lysis including suspending medium, the rate of freezing and thawing etc. (34, 35). Through further research, these parameters can be optimized to improve the extraction efficiency of biomolecules further and also to lyse bacterial cells with complex cell envelope.” Line 381-385

The manuscript also demonstrates that decreased cell viability during ULT-storage results in more efficient cell lysis. This can be an effect of the fraction of cells with a compromised membrane in the culture. In addition to MTT assay, performing propidium iodide (PI) staining in the cell culture would provide a better idea on quantifying the fraction of cells with such compromised membranes.

We appreciate this suggestion by the reviewer. This could be used to double confirmation our findings. So, we performed the PI assay and as expected, the results (Fig 1) complement those of MTT assay Figure 2 in the revised manuscript has results from the PI staining assay. Please take a look at the following sections (at the line numbers) for PI assay.

Materials and Methods: 

• Line 103-104, 

• Line 127-137.

Results and Discussion:

• Line 183-189, 

• Line 196-214, 

• Line 216-225, 

• Line 250-255.

For the comparison of graphs in figures 2 and 3, the authors have not mentioned what statistical tests were conducted and what the threshold for significance was set to be.

We thank the reviewer for bringing this to our notice. The suggestion has been incorporated in the revised figure (2 and 3) captions and materials and methods section.

Materials and methods: 

“Statistical analysis Mean values and standard deviations were calculated by Microsoft Excel standard functions. P-values used for determining statistical significance of our results were calculated in Microsoft Excel using Student’s t-test.” Line 176-179

Figure captions: 

Figures 1 and 2:

• “Data are shown as mean ± S.D, n =3.” Lines 224/ 242

Figures 3, 4 and 5:

• “The experiment was performed in biological triplicates. Statistical significance was determined using Student’s t-test (p<0.05). Data represented as Mean ± SD, n=3.” Lines 303-305/ 331-332

For the fluorescence protein procedure, the authors have used E. coli transformants expressing mCherry and eGFP. The manuscript does not provide the experimental details on generating these transformants.

We thank the reviewer for bringing this point to our notice. We borrowed these plasmid from Nielsen lab (acknowledgment provided) and we have included the appropriate plasmid names as well as the Biobrick ID (BBa_K2033011) & Addgene (Plasmid #75452) in the materials and methods section.

“E. coli BL21-DE3 transformants expressing mCherry fluorescent protein (BBa_K2033011 plasmid possessing ampicillin resistance; 100 µg/mL) and E. coli DH5α transformants expressing eGFP (pZE27GFP – Addgene plasmid #75452 possessing kanamycin resistance; 25 µg/mL), constitutively, were cultivated as above and used for estimating cell lysis efficiency (along with B. subtilis and Synechocystis 6803) and the impact of ULT storage on protein function.” Line 111-115

Finally, although the authors have pointed out various reasons that could have resulted in their observation, they have not been experimentally tested.

We appreciate the reviewer’s comment. However, we would like to clarify that the manuscript is primarily directed towards exploring the effect of simple ULT storage of harvested bacterial biomass upon further processing. As such, testing each of those variables would be beyond the scope of the study and so, we have provided appropriate citations to support our analysis. Few examples are provided below:. 

“Synechocystis 6803, on the other hand, has a unique extracellular matrix (S-layer) and multi-membrane (thylakoid) system within the cell that not only confers extended viability at -80℃, but is recalcitrant to disruption to extract the intracellular proteins (32). Hence, it can be corroborated that along with external physical stress and the presence of an outer membrane, intracellular organelle architecture, like thylakoid membranes may influence cell morphology, cell survival, cell lysis and therefore, protein extraction efficiency.” Line 289-294

“Apart from ULT storage time it is known that multiple other parameters have collaborative effect on bacterial cell lysis including suspending medium, the rate of freezing and thawing etc. (34, 35). Through further research, these parameters can be optimized to improve the extraction efficiency of biomolecules further and also to lyse bacterial cells with complex cell envelope.” Line 381-385

To further address your suggestion, we have also revised our introduction and conclusion to clearly convey our thoughts, which were little bit unclear in the original version. Please find the edits. 

Abstract:

“Ultra-low temperature (ULT) storage of microbial biomass is routinely practiced in biological laboratories.” Line 25-26

Introduction:

“Ultra-low temperature (ULT) storage of these recombinant cells and harvested biomass is a routine exercise in biological laboratories before subsequent sample processing; but its impact on the cells and proteins of interest has been minimally reported.” Line 57-60

Conclusion:

“Researchers often utilize ULT storage for storing the harvested microbial biomass during transport or for subsequent protein (or other biomolecules) extraction during the long-term experiment; but ignore its impact on the results of their subsequent experimentation.” Line 386-388

Considering your suggestion, we have removed certain supportive statements (as below) from the revised manuscript which were included earlier based on literature review:

….and structural integrity with respect to time and their composition.

… where the cell composition and metabolic regulation play important roles.

---

## [Decision Letter · Decision Letter 1]

30 Apr 2021

Novel perspective on a conventional technique: Impact of ultra-low temperature on bacterial viability and protein extraction

PONE-D-20-35156R1

Dear Dr. Varman,

We’re pleased to inform you that your manuscript has been judged scientifically suitable for publication and will be formally accepted for publication once it meets all outstanding technical requirements.

Kind regards,

Mehmet A Orman

Academic Editor

PLOS ONE

Additional Editor Comments (optional):

Reviewers' comments:

Reviewer's Responses to Questions

**Comments to the Author**

1. If the authors have adequately addressed your comments raised in a previous round of review and you feel that this manuscript is now acceptable for publication, you may indicate that here to bypass the “Comments to the Author” section, enter your conflict of interest statement in the “Confidential to Editor” section, and submit your "Accept" recommendation.

Reviewer #1: All comments have been addressed

Reviewer #2: All comments have been addressed

2. Is the manuscript technically sound, and do the data support the conclusions?

Reviewer #1: Yes

Reviewer #2: Yes

3. Has the statistical analysis been performed appropriately and rigorously? 

Reviewer #1: Yes

Reviewer #2: Yes

4. Have the authors made all data underlying the findings in their manuscript fully available?

Reviewer #1: Yes

Reviewer #2: Yes

5. Is the manuscript presented in an intelligible fashion and written in standard English?

Reviewer #1: Yes

Reviewer #2: Yes

6. Review Comments to the Author

Reviewer #1: The authors have done a good job in revising the manuscript and have included all the suggestions provided earlier.

Reviewer #2: The authors addressed the concerns that was raised on their initial submission. In my opinion, the manuscript reads better and is easier to follow. However, sentence "The fluorescence was obtained using spectrofluorometer..." (lines 134-136) sounds confusing. It would read better if this sentence is divided into two separate ones.

7. PLOS authors have the option to publish the peer review history of their article (what does this mean?). If published, this will include your full peer review and any attached files.

Reviewer #1: **Yes: **Amit Ghosh

Reviewer #2: No

---

## [Editor Report · Acceptance letter]

7 May 2021

PONE-D-20-35156R1 

Novel perspective on a conventional technique: Impact of ultra-low temperature on bacterial viability and protein extraction 

Dear Dr. Varman:

I'm pleased to inform you that your manuscript has been deemed suitable for publication in PLOS ONE. Congratulations! Your manuscript is now with our production department. 

Kind regards, 

on behalf of

Dr. Mehmet A Orman 

Academic Editor

PLOS ONE